# Prevalence of visual impairment and associated factors among children in Ethiopia: Systematic review and meta-analysis

Denekew Tenaw Anley[1]*, Rahel Mulatie Anteneh[1], Yibeltal Shitu Tegegne[2], Oshe Iemita Ferede[3], Melkamu Aderajew Zemene[1], Dessie Abebaw Angaw[3], Abraham Teym[4]

1 Department of Public Health, College of Health Sciences, Debre Tabor University, Debre Tabor, Ethiopia, 2 Amhara Regional Health Bureau, Bahir Dar, Ethiopia, 3 Department of Epidemiology and Biostatistics, Institute of Public Health, College of Medicine and Health Sciences, University of Gondar, Gondar, Ethiopia, 4 Department of Environmental Health, College of Health sciences, Debre Markos University, Debre Markos, Ethiopia

* denekewtenaw7@gmail.com

## Abstract

### Introduction

Visual impairment is a major public health problem in developing countries where there is no enough health-care service. It has a significant impact on the affected child's psychological, educational and socioeconomic experiences, during childhood and beyond. Therefore, the aim of this review was to estimate the pooled prevalence of visual impairment and its associated factors among children in Ethiopia.

### Method

This systematic review and meta-analysis was designed based on the PRISMA guidelines. Relevant published articles in Ethiopia from 2011–2021 were searched in PubMed/Medline, HINARI, Google scholar, and conference paper and thesis or research final reports were accessed from Ethiopian Universities' repositories. Data was extracted in Microsoft excel by using JBI data extraction checklist. The pooled prevalence and odds ratio of associated factors with their 95% CI was computed by using STATA 14/SE software. A fixed effect meta-analysis model was employed for a Cochrane Q test statistic and $I^2$ test showed there was no heterogeneity in the included studies.

### Result

A total of 7,647 children from nine studies were included in this study. The overall prevalence of visual impairment among children in Ethiopia was 7% (95% CI: 6, 7%). The pooled prevalence of visual impairment by region was almost similar in Ethiopia. However, there was no significant association between the identified factors and visual impairment among children. But the result showed that being males (AOR 0.642, 95% CI: 0.357–1.156), Children in the age of 10–13 years (AOR 0.224, 95% CI: 0.046–1.102) and 14–18 years (AOR

**Data Availability Statement:** The minimal anonymized data set necessary to replicate our

study findings is uploaded as Supporting Information file.

**Funding:** The author(s) received no specific funding for this work.

**Competing interests:** The authors have declared that no competing interests exist.

0.508, 95% CI: 0.102–2.534) were found to be less likely to have visual impairment. On the other hand, children of parents with visual impairment (AOR 1.820, 95% CI: 0.381–8.698) more likely to have visual impairment.

## Conclusion

Visual impairment among children in Ethiopia is still a public health problem one year later to VISION 2020, a global initiative aimed to eliminate avoidable blindness. All most one out of fourteen children in Ethiopia had visual impairment. Therefore, the government of Ethiopia should focus on effective, efficient, comprehensive eye health care services by integrating with the national health system to prevent avoidable visual impairment among children.

## Introduction

One of our most essential sensory systems and a mechanism of integration between the individual and the external environment is the visual system [1]. Visual impairment is significant loss of vison or functional limitation of the eye or the visual system. It can manifest as reduced visual acuity or contrast sensitivity, visual field loss, photophobia, diplopia, visual distortion, visual perceptual difficulties, or any combination of the above. Vision impairment ranges in severity from mild visual loss to total absence of light perception or blindness [2–4].

According to a recent international assessment on vision, at least 2.2 billion individuals worldwide have a vision impairment or blindness, with at least 1 billion of them having a visual impairment that could have been prevented or recognized but not handled [4].

Visual impairment and blindness are major public health problems in developing countries where there is no enough health-care service [5, 6]. Visual impairment among children with age 10 to 15 years old is more common in developing countries compared with developed one. According to the World Health Organization (WHO) 2010 reports, approximately 19 million children below 15 years of age are estimated to be visually impaired, while 1.4 million are blind based on WHO criteria. However, 80% of blindness is preventable [2, 7]. The prevalence of blindness in children ranges from approximately 0.3/1000 children in developed countries to 1.5/1000 in developing countries [8].

The prevalence of low vision in Ethiopia is 3.7% with considerable regional variation. The large proportion of this problem (91.2%) is due to avoidable (either preventable or treatable) causes. If it is early diagnosed and treated, the problem can be corrected easily. It could cause irreversible blindness otherwise [5]. According to findings to local areas, prevalence varies across regions, with the highest prevalence in Mikelle and the lowest prevalence in Gurage (12.4 percent and 5.20 percent, respectively) due to factors such as duration of mobile exposure, sex, and television distance varying from local to local [9, 10].

Globally, the most frequent causes of childhood visual impairment (both mild and sever) and blindness are retinal disorders, glaucoma, corneal scarring (primarily due to Vitamin A deficiency), cataract and cerebral cause. The other major causes of visual impairment are uncorrected refractive errors (43%) followed by cataract (33%); the first cause of blindness is cataract (51%) [6, 7]. Majority of the findings reported uncorrected refractive error as the major cause of visual impairment [11].

Visual impairment has a significant impact on the affected child's psychological, educational and socioeconomic experiences, during childhood and beyond [12, 13]. The control of childhood blindness is considered a high priority of the WHO's 'VISION 2020 with The

"Right to Sight' and "child eye health" as a public health agenda and used as a programme [14]. The main target of this global initiative was to eliminate avoidable blindness by the year 2020. Although Blindness in children is relatively uncommon, it was a priority of VISION 2020 for several reasons. The first one is children who are born blind or who become blind and survive have a lifetime of blindness, and ahead of them with all the associated emotional, social and economic costs to the child, family and society. The second one is that control of blindness is closely linked to child survival, as many of the conditions associated with childhood blindness also cause child mortality due to premature birth, measles, vitamin A deficiency [15, 16].

The evidence generated by this systematic review and meta-analysis could urge policy makers and program managers to design appropriate prevention, and detection strategies to reduce the risk of blindness and other negative consequences of visual impairment among children in Ethiopia. As to the best of our knowledge, there is no recent study on pooled estimate of the prevalence and associated factors of visual impairment among children in Ethiopia. Therefore, the aim of this systematic review and meta-analysis is to estimate the pooled prevalence of visual impairment and its associated factors among children in Ethiopia.

## Materials and methods

### Design and searching strategy

This systematic review and meta-analysis were done to compile the most recent evidences using articles published and grey literatures on the prevalence and associated factors of visual impairment among children in Ethiopia. The protocol was registered on PROSPERO international database with registration number of CRD42021233034. For reporting we followed the protocol of the Preferred Reporting Items for Systematic Review and Meta-Analysis (PRISMA) guideline [17].

Relevant published articles were searched in PubMed/Medline, HINARI, and Google scholar. In addition, other grey literatures were accessed from Ethiopian Universities' repositories. The search terms were developed in accordance with the Medical Subject Headings (MeSH) thesaurus using a combination of key terms (Visual impairment, children and Ethiopia) and then, the searching combination was adapted for use in other databases.

Manual searching of articles published in Ethiopian Journal of Health Sciences, Ethiopian Medical Journal, Ethiopian Journal of Health and Development, and Ethiopian Journal of Health and Biomedical Sciences was done. Reference lists of retrieved articles were traced to find out articles which were not retrieved from electronic databases using the developed searching combinations. Two author groups: group one (DT, RM) and group two (YS, OL), independently searched the articles. The searching of articles was done on January 24-25/2021 using the following searching combinations;

"Vision Disorders"[Title/Abstract] OR "Vision Disability "OR "Visual Disorders" OR" Visual Impairment"[Title/Abstract]) AND child [Title/Abstract] OR children [Title/Abstract] OR childhood [Title/Abstract] OR "school age children"[Title/Abstract])) AND (Ethiopia [Title/Abstract]).

The above searching strategy was inclusive of both institutional and community based studies as there was no a term used for exclusion.

### Inclusion and exclusion

Both descriptive and analytical cross-sectional studies were considered. Studies with prevalence and/or associated factors of visual impairment among children (0–18 years old) in Ethiopia, both community and institute-based studies with the outcome of interest and published in

English language from 2011–2021 were included. All citations without abstract and/or full-text and qualitative studies were excluded.

## Study selection and quality appraisal

All articles retrieved through search strategy were imported to EndNote X7. Then, duplications were checked and removed. After exclusion of duplicate studies, titles and abstracts were independently screened for inclusion in full text appraisal which were done by two groups of review authors: group one (DTA, RMA) and group two (YST, OL). Differences between two groups were resolved through discussion and /or the decision was determined by the third group of review authors (MAZ, DAA).

The full text of articles which we found relevant, were appraised for inclusion in systematic review and meta-analysis. The quality of studies was assessed using JBI critical appraisal checklist for prevalence studies and analytical cross-sectional studies having 9 and 8 checklist items, respectively [18, 19]. Articles with an overall quality assessment score of greater than half (50%) were included. The discrepancies during this full text quality assessment were solved like the way differences in title/abstract screening phase were resolved. The studies selection process was reported graphically using a PRISMA flow diagram [17].

## Outcome measurement

The main outcome of this systematic review and meta-analysis was childhood visual impairment in Ethiopia. The measure of effect was odds ratio. Low vision was defined as visual acuity of less than 6/18 but equal to or better than 3/60, or a corresponding visual field loss to less than 20˚, in the better eye with the best possible correction. Blindness was defined as visual acuity of less than 3/60, or a corresponding visual field loss to less than 10˚, in the better eye with the best possible correction. We defined vision impairment as vision worse than 6/12 in the better eye as it includes low vision (visual acuity worse than 6/18) and blindness (visual acuity worse than 3/60) [20].

## Data extraction

The data were extracted using JBI data extraction checklist. Two groups of review authors extracted the data independently. The differences between the two authors were solved with discussion. When there was no agreement, the decision was solved by the third authors review group. Information such as name of first author, year of publication, age group of study participants, study year, study area/region, study design, total number of participants, and number of visually impaired children or participants, proportion of visual impaired case and factors associated with visual impairment and measure of association for each factor in each study in Ethiopia were extracted using Microsoft excel spreadsheet.

## Data analysis

The extracted data were exported from Microsoft excel spreadsheet to STATA version 14 (SE) for analysis. Heterogeneity among included studies was quantitatively measured by index of heterogeneity ($I^2$ statistics), in which 25%, 50%, and 75% represented low, moderate, and high heterogeneity, respectively [21]. For the absence of heterogeneity, fixed effect model was used to estimate the pooled prevalence of visual impairment among children in Ethiopia. Because the studies are small, we have chosen fixed effect model, instead of random effect model even though the I statistics is elevated in determining the pooled effect size of associated factors [22]. Subgroup analysis was done by region to see the difference in the pooled prevalence of

visual impairment among regions. Small-study effect was evaluated using the visual funnel plot test, and Egger's test. Odds ratio with its 95% confidence was used to estimate the association between visual impairment and factors. The results were presented both in text and Forest plot.

## Results

### Searching results

Out of 103 articles retrieved, 30 studies were removed due to duplication through EndNote citation manager. Then, 58 studies were excluded after the title and abstract screening. Full publications of 15 articles were checked in detail for the presence of the outcome variable and 6 studies were removed. The remaining 9 eligible studies were included for this systematic review and meta-analysis to estimate the pooled prevalence of visual impairment after quality assessment using JBI quality assessment critical appraisal checklist.

From the total of 6 full text review article removed, 3 articles were excluded due to their outcome of interest was not directly related to our outcome of interest, visual impairment. And one article was excluded because of the study subjects were not similar to our study subjects. The remaining 2 articles were excluded for they were conducted outside Ethiopia. The overall study selection process was represented by the following flow diagram (**Fig 1**).

### Description of included studies

The characteristics of the 9 primary studies included in this review have been described in Table 1. Two Studies were descriptive cross sectional and 7 studies were analytical cross sectional studies carried out in different parts of Ethiopia having sample size in a range of 378 in Addis Ababa Arada sub city [23] to 1289 Gondar town in Amhara region [2]. These studies were conducted from 2011 to 2021.

In this meta-analysis, a total of 7,647 children were included to estimate the pooled prevalence of visual impairment. The 9 studies were conducted in different regions of the country: most of the studies were conducted in Amhara region and Addis Ababa [1, 2, 23–26], Tigray [9], Southern Nations, Nationalities and peoples' region (SNNPR) [10, 27]. The lowest and highest prevalence of visual impairment among children in Ethiopia was 4.4% [24] and 12.4%) [9] respectively. Independent evaluators re-assessed all the articles before any analysis and the studies were fit in terms of their quality (quality score ranged from 5 and above points). The description of included studies is presented by the following table (**Table 1**).

### Meta-analysis

In the estimation of pooled prevalence of visual impairment, nine studies were used and a total of 7,647 children were participated. The forest plot result of nine included studies showed that the overall pooled prevalence of visual impairment among children in Ethiopia was 7% (95% CI: 6, 7%) (Fig 2). As the $I^2$ statistics shows there was no heterogeneity. Hence, fixed effect model was used to estimate the overall pooled prevalence of visual impairment among children (**Fig 2**).

### The pooled prevalence of visual impairment by region

Subgroup analysis was done to see the pooled prevalence of visual impairment by region. According to the result, the highest prevalence was observed in Amhara region (7% (95% CI: 6, 8). The pooled prevalence was similar in Addis Ababa and SNNPR (6% (95% CI: 5, 7) (**Fig 3**).

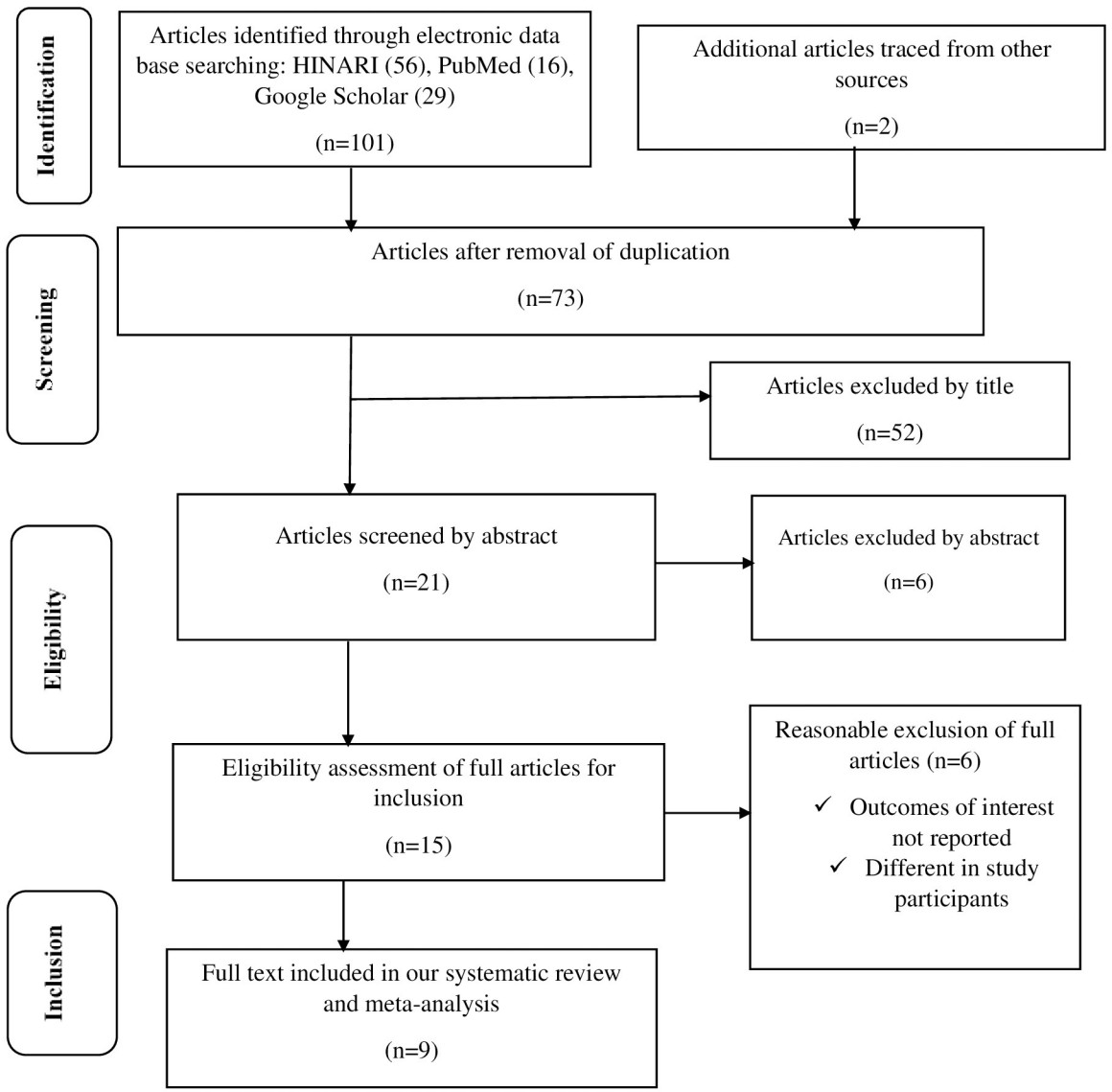

**Fig 1. Flow chart diagram describing selection of studies for the systematic review and meta-analysis of visual impairment and associated factors among children in Ethiopia, 2021.**

## Time trend of visual impairment

According to the line graph drawn, the prevalence of visual impairment had no fixed pattern. Publication in 2017 showed relatively higher prevalence of visual impairment in its study area. The prevalence was smaller in 2018 and decreased in 2020 (**Fig 4**).

## Determinants of visual impairment

Determinants of visual impairment were identified based on the pooled effect of two or more studies. The absence of significant heterogeneity was indicated by the insignificant p-values of $I^2$ in the estimate of pooled effect size of associated factors. However, there was no significant association between the identified factors and visual impairment among children. However,

**Table 1. Descriptive summary of 9 studies reporting the prevalence and associated factors of visual impairment among children in Ethiopia included in the systematic review and meta-analysis, 2021.**

| Author | Year of publication | Study design | Region | Study area | Sample size | Response rate | Prevalence (%) | Quality status |
|---|---|---|---|---|---|---|---|---|
| Alemu et al | 2014 | ACS | Amhara | Gondar town | 1289 | 100% | 5.43 | Low risk |
| Bezabih et al | 2017 | ACS | Addis Ababa | Addis Ababa | 804 | 89.3% | 7.24 | Low risk |
| Darge et al | 2017 | ACS | Addis Ababa | Arada sub city | 378 | 100% | 5.8 | Low risk |
| Dhanesha et al | 2018 | DCS | Tigray | Mekelle | 1197 | 95.1% | 12.4 | Low risk |
| Hailu et al | 2020 | ACS | Addis Ababa | Addis Ababa | 816 | 94.7% | 4.4 | Low risk |
| Kedir et al | 2014 | DCS | SNNPR | Gurage zone | 592 | 96% | 6.5 | Low risk |
| Merrie et al | 2019 | ACS | Amhara | Bahir dar | 632 | 95% | 8.7 | Low risk |
| Woldeamanuel et al | 2020 | ACS | SNNNPR | Gurage zone | 1064 | 100% | 5.2 | Low risk |
| Zelalem et al | 2019 | ACS | Amhara | Sekela woreda | 875 | 100% | 8% | Low risk |

Note: SNNPR: Southern Nations Nationalities and Peoples Region, Low risk: a study scored > 50% in the JBI quality assessment scale.

the result showed that males were less likely to have visual impairment compared to females with AOR 0.642(95% CI: (0.357–1.156). Children in the age of 10–13 years and 14–18 years were found to be less likely to have visual impairment compared to those in the age of 6–9

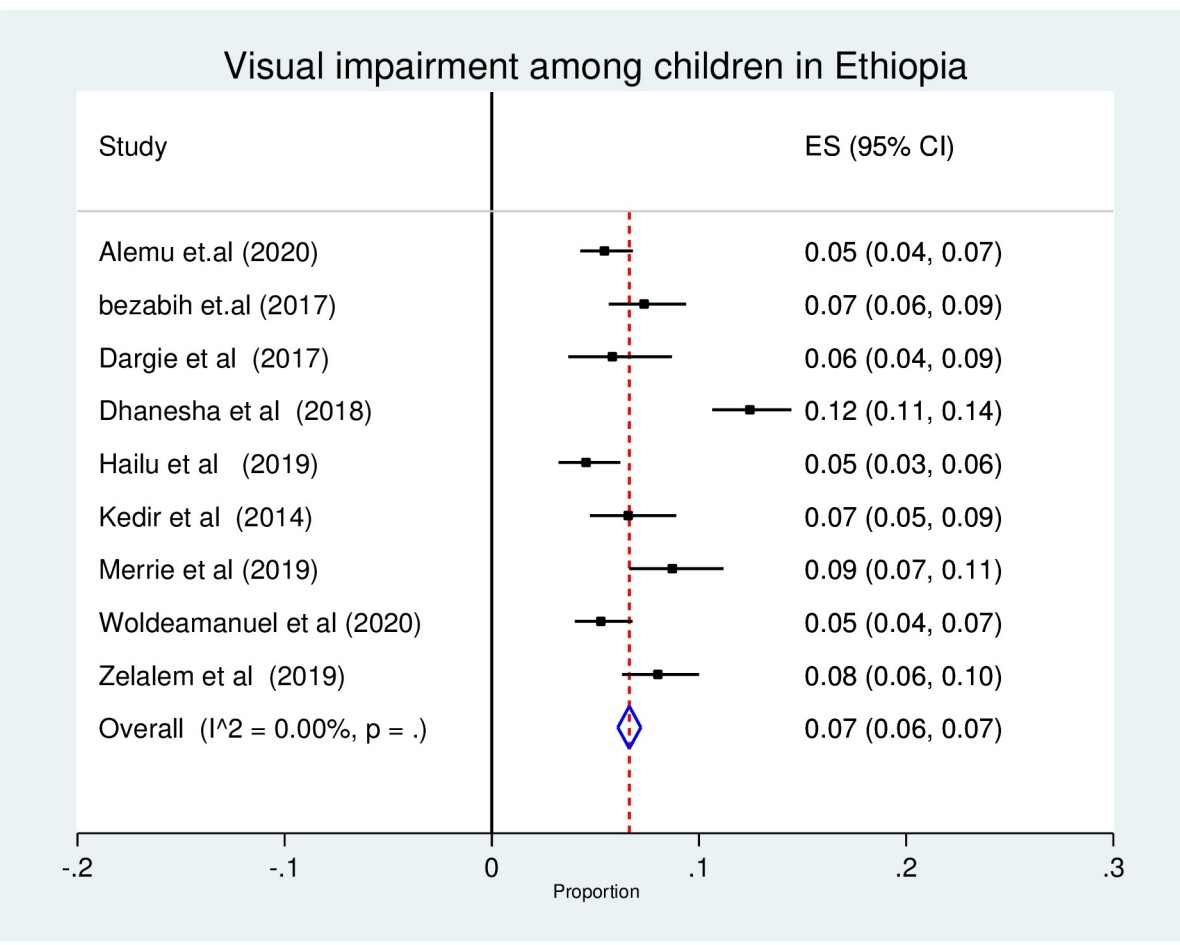

**Fig 2. Forest plot of the pooled prevalence of visual impairment among children in Ethiopia, 2021.**

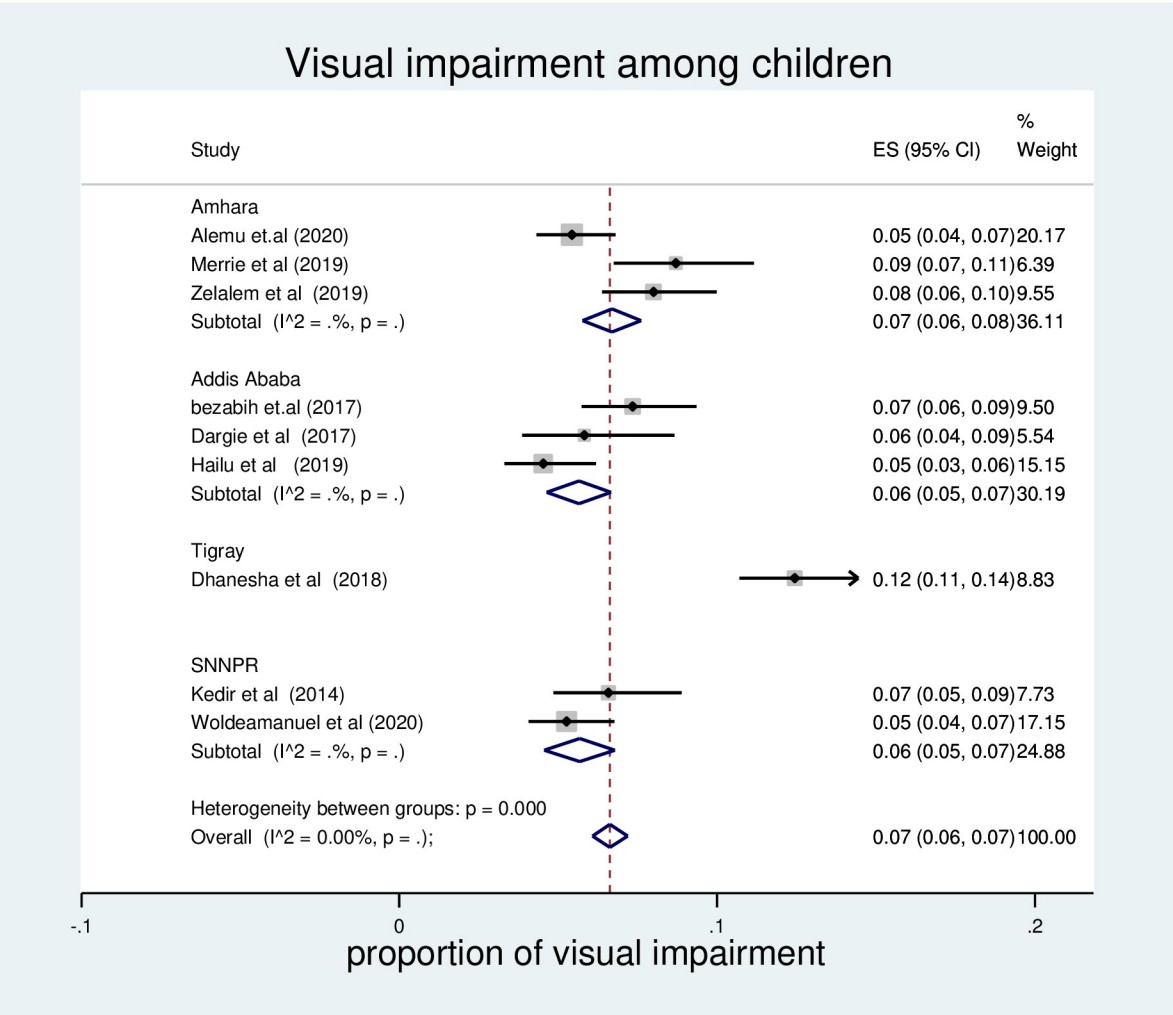

**Fig 3. Forest plot of pooled visual impairment among children in Ethiopia by region, 2021.**

years with AOR of 0.224 (95% CI; 0.046–1.102) and 0.508 (95% CI; 0.102–2.534) respectively. Children of families with visual impairment were 1.82 times more likely to have visual impairment compared to children of families with no visual impairment with AOR of 1.820 (95% CI; 0.381–8.698). Children from illiterate families were less likely to have visual impairment compared to those children of families who were college and above with AOR 0.668 (95% CI; 0.023–19.597) (**Table 2**).

## Small study effect

The presence of possible small study effect was checked by using funnel plot and egger test. The funnel plot showed symmetric distribution and the P-value for the egger's test was 0.117, both results indicated the absence of publication bias (**Fig 5**).

## Discussion

In this study, the pooled prevalence of visual impairment among children in Ethiopia was 7%. This finding is higher than the study conducted in china (0.66%) [28], a study conducted in

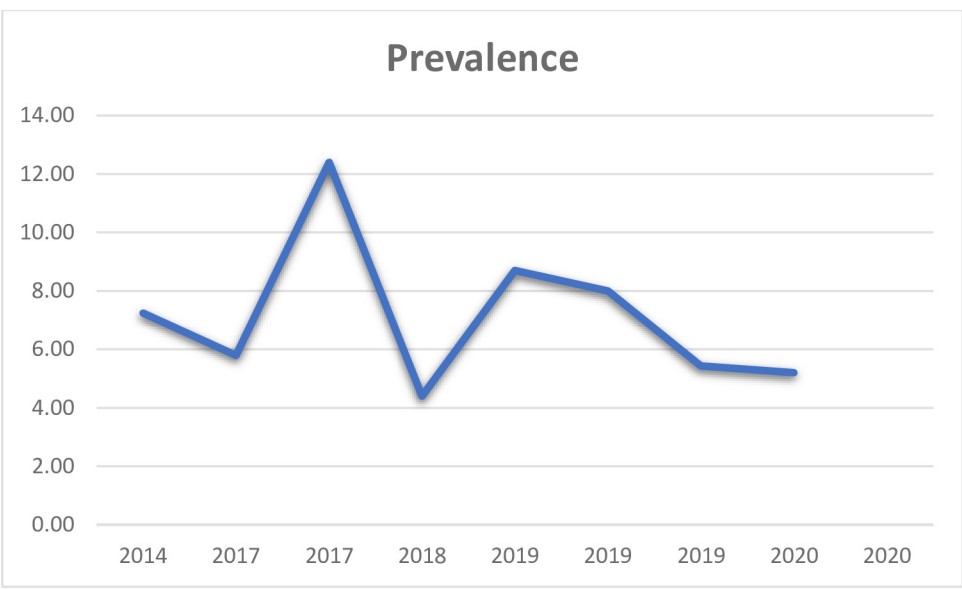

**Fig 4. Trend of visual impairment among children in Ethiopia, 2021.**

Australia (6.4%) [29], a study conducted in Brazil (4.82%) [30], but lower than the other study conducted in china (22.3%). This prevalence of visual impairment, higher than most of the studies, is because of high prevalence of trachoma among children in Ethiopia (26.9%) [31]. The other study also showed that prevalence of trachoma among children 1–9 years of age was 40.1% [32]. Studies showed that, trachoma was found to be the most common cause of visual impairment in Ethiopia [33]. The other reason can be the fact that Child eye health has not been given priority in health development plans of most African countries including Ethiopia [34]. The absence of national vision screening programs and surveillance data on visual impairment shows the existing fact of the absence of priority given to child health eye in Ethiopia. There is also hygiene problem particularly in rural areas of Ethiopia [35–37].

The result of sub group analysis by region showed that, the pooled prevalence of visual impairment was almost similar across regions in Ethiopia. However, there was relatively higher prevalence of visual impairment in Amhara region (7% 95%; CI; 6, 8) compared to Addis Ababa and SNNRP.

Regarding to determinants of visual impairment, none of the factors were found to be significantly associated. However, the effect size showed that males had lower odds of visual impairment compared to females (0.642 (0.357–1.156). This finding is also similar with other studies identified the higher odds of visual impairment like myopia among female children [38]. Children of parents with visual impairment were found to have higher odds of visual

**Table 2. The pooled effect size of factors of visual impairment among children in Ethiopia, 2021.**

| No. | Variable (Reference) | Number of studies | Effect size (95% CI) | Number of studies | Heterogeneity | | |
|---|---|---|---|---|---|---|---|
| | | | | | Q-value | P-value | $I^2$ |
| 1 | Sex (female) | 6 | 0.642 (0.357–1.156) | 6 | 0.81 | 0.976 | 61.0% |
| 2 | Age 10–13 (6–9) | 2 | 0.224 (0.046–1.102) | 2 | 0.25 | 0.616 | 80.1% |
| 3 | Age 14–18 (6–9) | 2 | 0.508 (0.102–2.534) | 2 | 0.05 | 0.828 | 80.1% |
| 4 | Illiterate parents (College and above) | 3 | 0.668 (0.023–19.597) | 3 | 0.06 | 0.970 | 72.9% |
| 5 | Parents with visual impairment (no visual impairment) | 2 | 1.820 (0.381–8.698) | 2 | 0.00 | 0.957 | 80.1% |

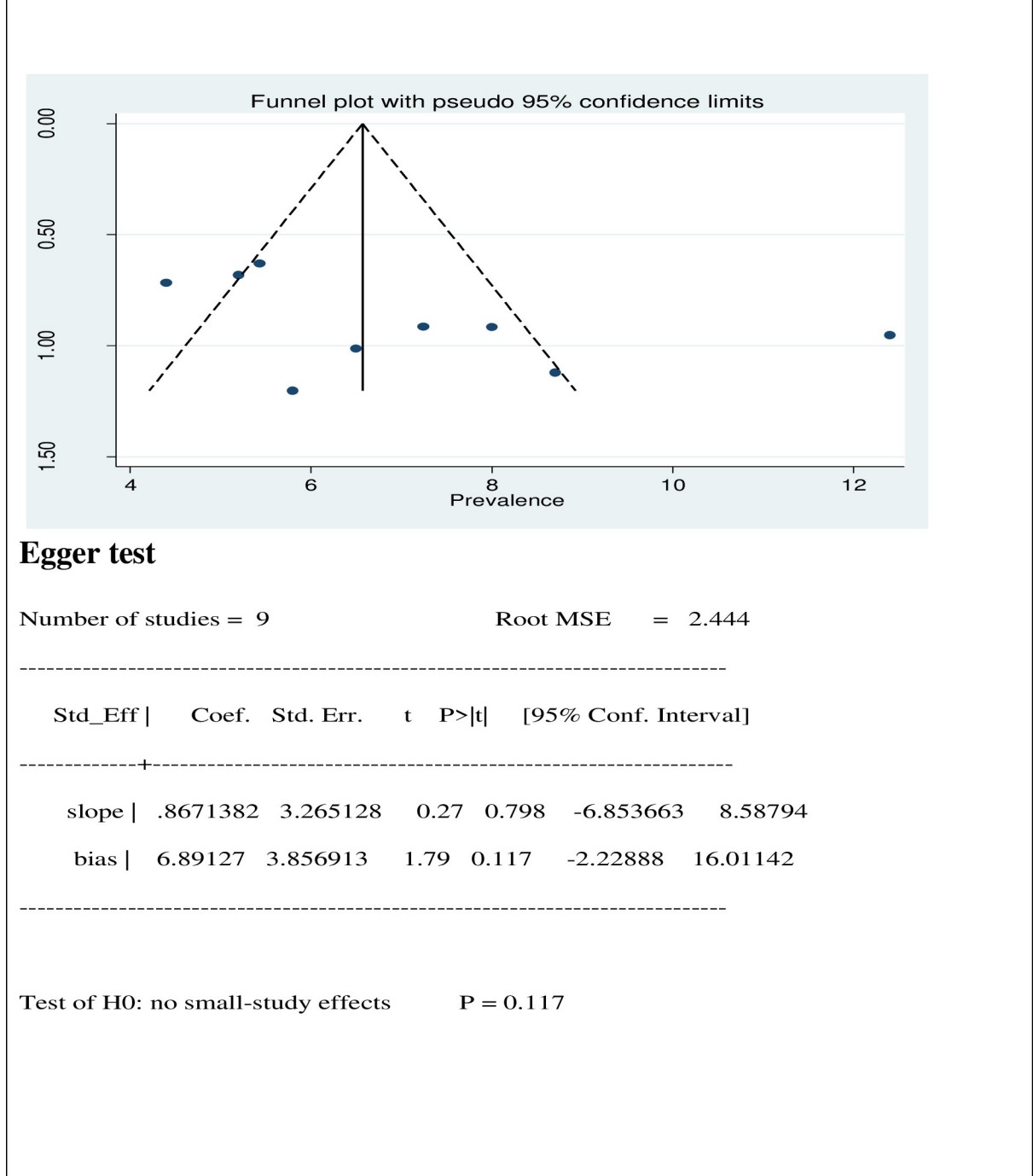

**Egger test**

Number of studies = 9                    Root MSE    = 2.444

--------------------------------------------------------------------------------

  Std_Eff |     Coef.   Std. Err.     t   P>|t|    [95% Conf. Interval]

-------------+------------------------------------------------------------------

    slope |   .8671382   3.265128    0.27   0.798   -6.853663     8.58794

    bias |   6.89127   3.856913    1.79   0.117    -2.22888    16.01142

--------------------------------------------------------------------------------

Test of H0: no small-study effects        P = 0.117

**Fig 5. Funnel plot and egger test of the 9 studies include in meta-analysis of visual impairment among children in Ethiopia, 2021.**

impairment (Pooled OR;1.820 (95% CI; 0.381–8.698)). The odds of visual impairment were found to be lower among children in the age of 10–13 and 14–18 years when compared to those in the age of 6–9 years. This may be due to the fact that children would take care of their eyes as they get matured.

### Strengths and limitations of the study

This study was the first systematic review and meta-analysis which showed the pooled prevalence of visual impairment among children based on the studies conducted in the last 10 years. However, the absence of studies from regions other than the included ones may limit the national representativeness of the study.

## Conclusion

Visual impairment among children in Ethiopia is still a public health problem one year later to VISION 2020, a global initiative aimed to eliminate avoidable blindness by the year 2020 and preventing the projected doubling of avoidable visual impairment between 1990 and 2020. All most one out of fourteen children in Ethiopia had visual impairment. In this review, there was no significant association between the identified factors and visual impairment among children. However, being male, children in the age of 10–13 years and 14–18 years were less likely to be visual impaired. Whereas, children of parents with visual impairment were more likely to have visual impairment. Therefore, the government of Ethiopia should focus on effective, efficient, comprehensive eye health care services by integrating with the national health system to prevent avoidable visual impairment among children.

## Supporting information

**S1 File. PRISMA 2020 checklist filled.**
(DOCX)

**S2 File. The minimal anonymized data set.**
(XLSX)

## Author Contributions

**Conceptualization:** Denekew Tenaw Anley, Rahel Mulatie Anteneh, Yibeltal Shitu Tegegne, Oshe lemita Ferede, Dessie Abebaw Angaw.

**Data curation:** Denekew Tenaw Anley, Dessie Abebaw Angaw.

**Formal analysis:** Denekew Tenaw Anley.

**Methodology:** Denekew Tenaw Anley, Rahel Mulatie Anteneh, Yibeltal Shitu Tegegne, Oshe lemita Ferede, Melkamu Aderajew Zemene, Abraham Teym.

**Visualization:** Melkamu Aderajew Zemene, Dessie Abebaw Angaw, Abraham Teym.

**Writing – original draft:** Denekew Tenaw Anley, Rahel Mulatie Anteneh, Oshe lemita Ferede, Melkamu Aderajew Zemene.

**Writing – review & editing:** Denekew Tenaw Anley, Yibeltal Shitu Tegegne, Melkamu Aderajew Zemene, Dessie Abebaw Angaw, Abraham Teym.

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
