## [Decision Letter · Decision Letter 0]

4 Apr 2022

PONE-D-21-27729

Prevalence of visual impairment and associated factors among children in Ethiopia: Systematic review and meta-analysis

PLOS ONE

Dear Dr. Anley,

Thank you for submitting your manuscript to PLOS ONE. After careful consideration, we feel that it has merit but does not fully meet PLOS ONE’s publication criteria as it currently stands. Therefore, we invite you to submit a revised version of the manuscript that addresses the points raised during the review process.

We look forward to receiving your revised manuscript.

Kind regards,

Ving Fai Chan, Ph.D., M.Sc., B.Optom.

Academic Editor

PLOS ONE

Journal Requirements:

2. Please provide the full electronic search strategy for at least one database, including any limits used, such that it could be repeated

5.  We noticed you have some minor occurrence of overlapping text with the following previous publication(s), which needs to be addressed:

- https://www.hindawi.com/journals/joph/2020/6934013/

- https://bjo.bmj.com/content/bjophthalmol/97/7/812.full.pdf

- https://bmcinfectdis.biomedcentral.com/articles/10.1186/s12879-019-4686-8

In your revision ensure you cite all your sources (including your own works), and quote or rephrase any duplicated text outside the methods section. Further consideration is dependent on these concerns being addressed.

Additional Editor Comments (if provided):

Dear Dr Anley,

This paper has been reviewed and the reviewers highlight a number of of strengths and weaknesses.

I could see the merit in this manuscript. However, Reviewer 1 has pointed a few critical points that needs addressing for a systematic review (title, clear research questions, search strategy, management of heterogeneity, etc) before this manuscript is fit to be further consideration. Please pay close attention to the comments. Some may require major changes. Furthermore, there is a need for overall language editing (as pointed out by Reviewer 2). I believe that once these are addressed, the manuscript could be resubmitted for publication.

Regards,

Dr Ving Fai Chan

Reviewers' comments:

Reviewer's Responses to Questions

**Comments to the Author**

1. Is the manuscript technically sound, and do the data support the conclusions?

Reviewer #1: Yes

Reviewer #2: Yes

2. Has the statistical analysis been performed appropriately and rigorously? 

Reviewer #1: Yes

Reviewer #2: Yes

3. Have the authors made all data underlying the findings in their manuscript fully available?

Reviewer #1: Yes

Reviewer #2: Yes

4. Is the manuscript presented in an intelligible fashion and written in standard English?

Reviewer #1: Yes

Reviewer #2: No

5. Review Comments to the Author

Reviewer #1: 1.The purpose of this systematic review is to report pooled estimate of the prevalence and associated factors of visual impairment among children in Ethiopia. However the “Determinants of Visual impairment “ it covered were not of interest. As the author had discussed in the article, the cause of visual impairment would be of more clinical interest and value.

2. Literature Searches and Search terms are incomplete. This is suboptimal for publication for systematic review. Search terms should better include any population based study on vision, not only for child. Those studies would include a group of patients in this age range.

3. There is substantive heterogeneity in estimating pooled effect size of factors of visual impairment among children. Please explain the way of management in the method , result and discussion.

4. What is the exact criteria for low vision. In the outcome measurement, line 173 less than 1/18, line 177 less than 6/12.

Also some writing mistakes like in line 107 met analysis.

Reviewer #2: General

The article ‘Prevalence of visual impairment and associated factors among children in Ethiopia:

Systematic review and meta-analysis’ is very interesting and provide useful information to readers.

The manuscript contains original findings.

However, there are few issues need to be consider.

Title Page

The title is appropriate.

Th name of institution should be written in capital letter (page 1 line 5 and line 11).

Abstract

The abstract summarizes clearly and concisely the main finding of the results.

Main Manuscript

1. Introduction

The introduction of the study is appropriate.

Need to rephrase sentence in line 91 and line 92 (page 3).

Paragraph 3 (page 4) is not clear. Need to rephrase/require English editing.

Should delete the abbreviation VI (for visual impairment) (page 4 line 95) and BL (for blindness) (page 4 line 97) since those abbreviation are not consistently used in the text.

2. Methods

The methods give enough detail.

The name of journal should be written in capital letter (page 5 line 137 and line 138).

Need to rephrase sentence in line 142 (page 5).

Should delete the abbreviation VA (for visual acuity) (page 6 line 177) since this abbreviation is not consistently used in the text.

3. Results

The results are presented in a clear and concise manner.

4. Discussion

The discussion interprets the findings based on the results obtained and compare with previous studies for the prevalence of visual impairment.

However, for the associated factors for visual impairment, there is lacking comparison with other studies.

5. Conclusion

The conclusions are valid and based on the results of the study.

References

There are missing name of journal in few number of references.

Figures

The figures and figure legend are appropriate, clear and correctly labelled.

Tables

The tables and table legend are appropriate, clear and correctly labelled.

6. PLOS authors have the option to publish the peer review history of their article (what does this mean?). If published, this will include your full peer review and any attached files.

Reviewer #1: No

Reviewer #2: No

---

## [Author Response · Author response to Decision Letter 0]

18 Apr 2022

Thank you for your constructive comments. We have responded to all comments point by point. The point by point response letter is uploaded with the revised manuscript files.

---

## [Decision Letter · Decision Letter 1]

17 May 2022

PONE-D-21-27729R1Prevalence of visual impairment and associated factors among children in Ethiopia: Systematic review and meta-analysisPLOS ONE

Dear Dr Denekew Tenaw Anley,

Thank you for submitting your manuscript to PLOS ONE. After careful consideration, we feel that it has merit but does not fully meet PLOS ONE’s publication criteria as it currently stands. Therefore, we invite you to submit a revised version of the manuscript that addresses the points raised during the review process.

We look forward to receiving your revised manuscript.

Kind regards,

Ving Fai Chan, Ph.D., M.Sc., B.Optom.

Academic Editor

PLOS ONE

Additional Editor Comments:

Dear Dr Denekew Tenaw Anley,

After reviewing the responses to the queries raised by both reviewers, there are still critical issues that need addressing. Please pay attention to Reviewer 1's comments on the principles of conducting of systematic reviewers - research questions, search strategy, definitions and heterogeneity (with its impact on the interpretation of results). I felt that this manuscript has significant public health value, and will offer your team a second revision.

Regards,

Dr Ving Fai Chan

Reviewers' comments:

Reviewer's Responses to Questions

**Comments to the Author**

1. If the authors have adequately addressed your comments raised in a previous round of review and you feel that this manuscript is now acceptable for publication, you may indicate that here to bypass the “Comments to the Author” section, enter your conflict of interest statement in the “Confidential to Editor” section, and submit your "Accept" recommendation.

Reviewer #1: (No Response)

Reviewer #2: All comments have been addressed

2. Is the manuscript technically sound, and do the data support the conclusions?

Reviewer #1: (No Response)

Reviewer #2: Yes

3. Has the statistical analysis been performed appropriately and rigorously? 

Reviewer #1: (No Response)

Reviewer #2: Yes

4. Have the authors made all data underlying the findings in their manuscript fully available?

Reviewer #1: (No Response)

Reviewer #2: Yes

5. Is the manuscript presented in an intelligible fashion and written in standard English?

Reviewer #1: (No Response)

Reviewer #2: Yes

6. Review Comments to the Author

Reviewer #1: The answers were not fully satisfying, as follows:

The associated factors included in this systematic review and meta analysis are those identified by the articles included in this study.

Yes, this is the inherent limitation of this study. As the “key question” is “the Prevalence of visual impairment”, it is acceptable. However the value of this systematic review may be limited.

The study was supposed to include both institutional and community based studies, and hence there was no any exclusion made in searching strategy.

My concern is there might be other population based studies which include child. The searching combination " child [Title/Abstract] OR children [Title/Abstract] OR childhood [Title/Abstract] OR "school age children"[Title/Abstract] might exclude these studies.

However, the p-value for the identified I2 values in estimating pooled effect size of factors were found to be insignificant for all factors.

My understanding from table 2, all the I2 indexes were larger than 30%. It indicates the variability in the measured effect sizes across studies is caused by true heterogeneity among studies. Though a nonsignificant value of Q statistics (not I2) suggests that the studies are homogeneous, the Q statistic has limited power to detect heterogeneity in meta-analyses with few studies (ie 2-6 in this study).

Outcomes defined were “low vision”, “blindness”, and “visual impairment”. The outcome of interest for this study is “visual impairment” which can include both low vision and blindness. It is more inclusive than the two aforementioned outcomes. Hence, the last visual acuity, 6/12, is the one which defines visual impairment. Other respective values define low vision and blindness in that order.

This is hard to understand. If visual impairment includes both low vision and blindness, why 6/12 not 6/18?

Reviewer #2: The authors have adequately addressed my comments.

General

The article ‘Prevalence of visual impairment and associated factors among children in Ethiopia:

Systematic review and meta-analysis’ is very interesting and provide useful information to readers.

The manuscript contains original findings.

Title Page

The title is appropriate.

Abstract

The abstract summarizes clearly and concisely the main finding of the results.

Main Manuscript

1. Introduction

The introduction of the study is appropriate.

2. Methods

The methods give enough detail.

3. Results

The results are presented in a clear and concise manner.

4. Discussion

The discussion interprets the findings based on the results obtained and compare with previous studies for the prevalence of visual impairment.

5. Conclusion

The conclusions are valid and based on the results of the study.

Figures

The figures and figure legend are appropriate, clear and correctly labelled.

Tables

The tables and table legend are appropriate, clear and correctly labelled.

7. PLOS authors have the option to publish the peer review history of their article (what does this mean?). If published, this will include your full peer review and any attached files.

Reviewer #1: No

Reviewer #2: No

---

## [Author Response · Author response to Decision Letter 1]

10 Jun 2022

A rebuttal letter 2

Journal name: PLOS ONE 

PONE-D-21-27729

Title: Prevalence of visual impairment and associated factors among children in Ethiopia: Systematic review and meta-analysis

Thank you all for giving us your valuable time. The Editor’s, reviewers’ comments, and point by point response of the authors are presented by the following table. 

Comments from reviewer #1 Authors’ response

1. The study was supposed to include both institutional and community based studies, and hence there was no any exclusion made in searching strategy.

My concern is there might be other population based studies which include child. The searching combination " child [Title/Abstract] OR children [Title/Abstract] OR childhood [Title/Abstract] OR "school age children"[Title/Abstract] might exclude these studies. Thank you for your comment. As we have said before, there was no any exclusion made in the searching strategy regarding to institutions where the studies might be conducted. Therefore, there was no way that community based studies on visual impairment among children would be missed. 

2. My understanding from table 2, all the I2 indexes were larger than 30%. It indicates the variability in the measured effect sizes across studies is caused by true heterogeneity among studies. Though a non-significant value of Q statistics (not I2) suggests that the studies are homogeneous, the Q statistic has limited power to detect heterogeneity in meta-analyses with few studies (ie 2-6 in this study).

 Thank you reviewer for your valuable comment. The insignificant p-values of Q statistics assured us the absence of true heterogeneity in the effect sizes even though it has limited power. We preferred not to use random effect model even though the I statistics is elevated, for it is not advisable to use it when studies are small. 

3. Outcomes defined were “low vision”, “blindness”, and “visual impairment”. The outcome of interest for this study is “visual impairment” which can include both low vision and blindness. It is more inclusive than the two aforementioned outcomes. Hence, the last visual acuity, 6/12, is the one which defines visual impairment. Other respective values define low vision and blindness in that order.

This is hard to understand. If visual impairment includes both low vision and blindness, why 6/12 not 6/18?

 Thank you for this important comment. Something which has to be clear here is that our outcome of interest is “visual impairment” of any extent. We can forget about “low vision” and “Blindness”. The maximum visual acuity measurement value which defines visual impairment is 6/12. All results of visual acuity measurement bellow 6/12 will also indicate the presence of visual impairment. As you have said, visual acuity measurement 6/18 also indicates visual impairment, but it is not the maximum limit which defines the outcome. 

Comment from Reviewer #2: 

The authors have adequately addressed my comments. Thank you dear reviewer, for giving us your valuable time for the improvement of our manuscript.

---

## [Editor Report · Decision Letter 2]

15 Jun 2022

PONE-D-21-27729R2Prevalence of visual impairment and associated factors among children in Ethiopia: Systematic review and meta-analysisPLOS ONE

Dear Dr. Anley,

Thank you for submitting your manuscript to PLOS ONE. After careful consideration, we feel that it has merit but does not fully meet PLOS ONE’s publication criteria as it currently stands. Therefore, we invite you to submit a revised version of the manuscript that addresses the points raised during the review process.

Thank you for your responses to Reviewer 1. I suggest that you make a few minor edits in your manuscript so that readers not from the eye field can understand the subject matter better. For example:

a. You can include a line in your Methods, highlighting that "the search strategy included both institutional and community-based studies". The aim is to ensure your readers understand that the robustness of your search strategy.

b. Also in Methods, highlight that "we have chosen X, instead of random effect model, even though the I statistics is elevated because the studies are small (cite)." The aim is to ensure your readers understand the rationale of your choice of analysis.

c. Also in Methods, highlight that "we defined vision impairment as vision worse than 6/12 in the better eye as it includes low vision (VA definition) and blindness (VA definition)".

I hope this helps.

We look forward to receiving your revised manuscript.

Kind regards,

Ving Fai Chan, Ph.D., M.Sc., B.Optom.

Academic Editor

PLOS ONE
---

## [Author Response · Author response to Decision Letter 2]

29 Jun 2022

A rebuttal letter 3

Journal name: PLOS ONE 

PONE-D-21-27729

Title: Prevalence of visual impairment and associated factors among children in Ethiopia: Systematic review and meta-analysis

Thank you for giving us your valuable time. The Editor’s comments and point by point responses of the authors are presented by the following table. 

Comments from the editor Authors’ response

1. Thank you for your responses to Reviewer 1. I suggest that you make a few minor edits in your manuscript so that readers not from the eye field can understand the subject matter better. For example:

a. You can include a line in your Methods, highlighting that "the search strategy included both institutional and community-based studies". The aim is to ensure your readers understand that the robustness of your search strategy.

b. Also in Methods, highlight that "we have chosen X, instead of random effect model, even though the I statistics is elevated because the studies are small (cite)." The aim is to ensure your readers understand the rationale of your choice of analysis.

c. Also in Methods, highlight that "we defined vision impairment as vision worse than 6/12 in the better eye as it includes low vision (VA definition) and blindness (VA definition)".

 Thank you for your valuable comments. As per your comment, we have highlighted each responses mentioned “a to c” in the methods and materials section of the revised manuscript. The changes made are also indicated in the revised manuscript with track changes.

2. Please review your reference list to ensure that it is complete and correct. If you have cited papers that have been retracted, please include the rationale for doing so in the manuscript text, or remove these references and replace them with relevant current references. Any changes to the reference list should be mentioned in the rebuttal letter that accompanies your revised manuscript. If you need to cite a retracted article, indicate the article’s retracted status in the References list and also include a citation and full reference for the retraction notice. Thank you for your valuable comment. As per your recommendation, we have reviewed the references for completeness and correctness. We found that all are correct and complete. No retracted papers are cited.

---

## [Editor Report · Decision Letter 3]

1 Jul 2022

Prevalence of visual impairment and associated factors among children in Ethiopia: Systematic review and meta-analysis

PONE-D-21-27729R3

Dear Dr. Anley,

We’re pleased to inform you that your manuscript has been judged scientifically suitable for publication and will be formally accepted for publication once it meets all outstanding technical requirements.

Kind regards,

Ving Fai Chan, Ph.D., M.Sc., B.Optom.

Academic Editor

PLOS ONE

---

## [Editor Report · Acceptance letter]

11 Jul 2022

PONE-D-21-27729R3 

Prevalence of visual impairment and associated factors among children in Ethiopia: Systematic review and meta-analysis 

Dear Dr. Anley:

I'm pleased to inform you that your manuscript has been deemed suitable for publication in PLOS ONE. Congratulations! Your manuscript is now with our production department. 

Kind regards, 

on behalf of

Dr Ving Fai Chan 

Academic Editor

PLOS ONE